# An assessment of critical thinking in the Middle East: Evaluating the effectiveness of special courses interventions

**Wael Yousef** *

Core Curriculum Program, Qatar University, Doha, Qatar

* wyousef@qu.edu.qa

**Data Availability Statement:** All relevant data are within the manuscript and its Supporting Information files.

## Abstract

Critical thinking is a requisite skill for college success, employability, and conducive active civic participation. Empirical studies have noted to the low achievement of Arab students on critical thinking assessments. Insufficient endeavors have attempted to propose effective interventions enhancing critical thinking abilities among Arab students. The current analysis provides a preliminary overview of a special course designed to improve critical thinking skills among Arab college students. Results indicated a great improvement in all areas of critical thinking including explanation of information, identification of strategies, implementing solutions, and formulating logical inferences. Students' scores on a critical thinking assessment increased from sufficient to good as a result of participating in the program. The gains are consistent after controlling for gender, major, class seniority, and nationality. Notwithstanding these promising results, this paper is limited in several respects including the choice of critical thinking assessments represented by two questions, the highly contextualized setting making it difficult to be replicated, and the convenient sampling strategy used to recruit participants. This set of limitations, however, does not discourage proactive attempts like designing special courses to enhance students' critical thinking acquisition in the Middle East.

## Introduction

Critical thinking is reflective judgment involving the explanation, evaluation, analysis, and interpretation of information to generate logical inferences [1]. Critical thinking improves students' academic achievement, career outcomes, and civic engagement [2, 3]. The study of critical thinking in Arab countries suffers from several conceptual and methodological problems warranting systematic investigation [4, 5].

First, researchers have utilized translated American and European assessments without proper modifications [6]. Second, studies focused on a single course, college, or program [7]. Third, proposed interventions for enhancing critical thinking skills were designed for small homogenous populations [8]. Fourth, published critical thinking skills interventions lacked details on the treatment, instructional strategies, assessment tools, or teaching styles [9]. Fifth,

**Funding:** The author received no specific funding for this work.

**Competing interests:** The author has declared that no competing interests exist.

interventions were not grounded in well-established educational theories [7–9]. Sixth, studies failed to include adequately large or representative samples of populations under investigation [6]. Seventh, experimental, and quasi-experimental studies featured little to no control over potential confounders [7].

This manuscript offers several contributions to the literature on critical thinking among Arab college students. First, it provides an exploration of critical thinking skills of Qatari students critical thinking abilities. This evaluation of critical thinking is no way comprehensive nor systematic. It provides a glimpse of Qatari students' critical thinking abilities informing future interventions to further enhance this quality among college students. Second, it proposes an intervention that could be further developed with the necessary details to be scalable and validated with larger random samples of students. This intervention was highly contextual given the different instructors, activities, tasks, and discussions involved in each class session making it difficult to be replicated. Nevertheless, the present manuscript provides an overview of the basic components of the intervention.

Third, the study presents the findings from a validation study of the critical thinking instrument used. Note, however, that the instrument was made of two questions, which is limiting. Further, the sample of the study was convenient, which also presents legimate concerns about the reliability and validity of scores generated. Despite such limits, the paper provides a framework of how future researchers interested in critical thinking assessments in the Arab World need to validate their assessments. Fourth, the current manuscript provides promising results on the effectiveness of interactive teaching methods to foster critical thinking skills. While the exact activities, tasks, and discussions differed from a session to another throughout the course, one fundamental defining feature of the course was increased interactivity, engagement, and collaboration between all participants in their class. This provides a moving ground for future researchers to design specific activities enhancing interactivity to be used in critical thinking courses.

## Literature review

### Conceptualizing critical thinking

There is a plethora of critical thinking conceptualizations [2, 9]. Many refer to critical thinking as transferrable, cross-disciplinary, and universal set of dispositions and skills [5]. This collection of skills pertains to the ability of identifying relevant information, problem solving strategies, logical reasoning, and using evidence to construct and defend arguments [6]. Others understand critical thinking as domain-specific [7]. This analysis views critical thinking as a universal set of practices applicable across disciplines [8]. The view of the current investigation is that critical thinking could be taught at the college level, and its skills are transferable to various academic, as well as, non-academic applications [9].

There are several systematic differences between the assessment of Arab students on critical thinking compared to many students' populations like those of North America, Europe, and East Asia. First, Arab students are exposed to less frequent, as well as systematic critical thinking appraisals. Second, Arab colleges and universities seldom evaluate their students' abilities in critical thinking. Third, courses at both the undergradute and graduate levels in Arab colleges and universities do not feature high levels of cognitively demanding tasks like intensive writing, experiential projects, in-class simulations, or independent research. Fourth, Arab college students have performed systematically lower on critical thinking assessments compared to their counterparts around the world. Fifth, Arab students are assessed using translated non-authentically developed instruments within their own environments leading to biased understandings of several items on critical thinking appraisals.

## Assessment of critical thinking skills among Arab college students

A reading of the empirical literature on critical thinking skills among Arab college students points to a weak ability in the various populations observed by researchers across different settings [10–13]. Salih [10] found low aptitude levels among a large sample of Iraqi students in the areas of information organization, arguments construction, and using empirical evidence in constructing explanations. Similarly, Al-Hadabi and Al-Ashwal [11] reported low abilities in the identification of relevant information, writing coherent arguments, and justifying constructed arguments among a large sample of Yemeni students. Al-Ziq [12] found that Jordanian students exhibit insufficient abilities in recognizing patterns, themes, and typologies when presented with complex sets of information. By the same token, Al-Bjaidi [13] found pervasive inconsistencies in female Saudi students' abilities in identifying significant, relevant, and crucial information in independent research projects. Al-Mahrooqi and Denman [14] concluded that Omani students at the largest research institution in the country possess unacceptable skills in categorizing, classifying, and meaningfully coding similar information.

Assessments of critical thinking abilities among Arab college students have noted weak research skills [15, 16]. Arab students across countries and domains have been consistently found to lack skills in questions and hypothesis formulation [17, 18]. Further, Arab college students have developed inadequate data analysis, collection, and organization abilities [19, 20]. By the same token, studies have concluded a noticeable deficiency in coherent organization of ideas, evidence, and arguments in assignments requiring extensive writing. Authors have blamed traditional instructor-centered teaching methods focusing on memorization rather than critical reasoning and problem solving [21, 22].

## Instructional methods fostering critical thinking

Collaborative, participatory, experiential, engaging, and independent learning instructional strategies have fostered improved levels of critical thinking abilities among college students [23, 24]. Further, the use of deliberative and reflective assessments has enhanced students' critical thinking skills and dispositions [25, 26]. Embedding real-world illustrations and case-based teaching allows students to practice intended critical thinking skills [27].

The empirical evidence supports the use of participatory, engaging, collaborative, and reflective instruction to foster higher levels of critical thinking [28–32]. Lee, Lee, Gong, Bae, and Choi [28] reported standardized mean differences of 0.42 and 0.29 in favor of non-lecturing methods compared to lecturing techniques on the California Critical Thinking Dispositions Inventory (CCTDI) and the California Critical Thinking Skills Test (CCTST) in their meta-analysis [28]. By the same token, Miterianifa, Trisnayanti, Khoiri, and Ayu [29] concluded that problem-based learning in science courses improved critical thinking abilities significantly by reporting large effect sizes of 1.2 (standardized means' difference) in their meta-analysis. Abrami, Bernard, Borokhovski, Waddington, Wade, and Persson [30] concluded that exposure to real-world problems, collaborative, and deliberative learning, and mentoring all significantly increase students' appreciation and mastery of critical thinking in an extensive meta-analysis. Chen, Tseng, and Hsiao [31] performed a meta-analysis on digital games' effects on language learning, reporting moderate to substantial gains students accrue from using adventurous, challenging games in their vocabulary retention, reading comprehension, and ability to connect varying pieces of evidence to formulate a clear linguistic argument. Qian and Clark [32] reviewed 29 empirical articles investigating the effects of games on 21st-century skills, critical thinking dispositions, and abilities, reporting moderate, consistent positive associations between varying games' features (challenge, strategies identification, implementation of techniques, and discovery) and students' mastery of critical thinking skills.

## Research design

This study is a mixed-methods research utilizing experimental, as well as focus groups methods. The experimental design investigates the effectiveness of a proposed course on a sample of Qatar University critical thinking abilities. The focus group collects information on students' thoughts about the utilized instructional strategies during their course, as well as their conceptions of their thoughts processes progression.

## The quantitative phase

**Pre-test and post-test one group quasi-experimental design.**   A pre-posttest one group design was selected to investigate the effectiveness of a critical thinking course on students' abilities at Qatar University. This designs features pre-testing and post-testing participants on the same instrument, a critical thinking test. Participants receive a treatment, which is a special quantitative reasoning course between the two measurements on the test. All participants are considered to be one group with no control subjects. This is a prevalent design in education research investigating the effectiveness of new courses, teaching methods or instructional techniques.

Administrative, practical, and ethical concerns informed the choice for the one group pretest post-test quasi-experimental design. First, the university mandated all instructors to teach the same curriculum in their classes, and any changes to the content, teaching strategies, and assessment methods will not be honored. Second, on an ethical note, different curriculum, instruction, or assessment is associated with varied students' learning outcomes. To guarantee similar levels of educational outputs, the entire group of students received the same course. Third, a control group requires a different treatment, whether the old traditional course or another choice, which was difficult to envisage given the university's shrinking budget, resources, and personnel. For such considerations, the pre-test, post-test one-group design was selected to perform the analysis.

**Participants.**   Eighty students attending Qatar University in the Spring of 2018 participated in this research. Students took the course as a required subject by the university; however changes were made to the syllabus reflecting its intent to improve their critical thinking skills. They majored in liberal arts, business, technology, science, and law. All participants were between the ages of 18 and 25. 72% of the participants were females. 63% of the participants were Qatari citizens. Only 30% of participants reported working either part-time or full-time. 35% were freshmen, 28% were sophomore, 26% were juniors, and 11% were seniors. 82% of participants reported speaking Arabic fluently, while 71% indicated their English-proficiency to be good or excellent. Students came from 11 different countries. The student attended seven different sections of the same course taught by three instructors who were full-time tenured faculty.

**Measures.**   The outcome variable is critical thinking ability. Two questions assessed it. Each of the questions was graded on five measures: the student's ability to explain the issues, the skill of identifying a strategy, the extent to which the strategy was implemented, and the overall inference. The fifth measure was an average score of the four abilities, as mentioned above. Each measure was scored on a 1–4 rubric where 1 = poor, 2 = fair, 3 = good and 4 = excellent. The criteria used to assess each students' response is represented in the rubric included in S1 Appendix. The translation of the two questions used is presented in S2 Appendix. Note that all answers were published with the permission of students.

The two questions' instrument intended to measure four fundamental dimensions of critical thinking ability: identification of relevant information, the formulation of a strategy, the implementation of the chosen strategy, and inferring an interpretable statement based on the

evidence available. Each of the question requires learners to engage in all four conceptual cornerstones of critical thinking. The use of two questions allows the assessment of critical thinking in a rapid manner without overburdening students with taking hours-long assessments. Correlations between students' scores on the two questions and a popular appraisal yielded high associations signaling the validity of the two questions' instrument.

The main independent variable or the treatment was a math course called mathematics in society. The course consists of fifteen modules, each geared to improve students' critical thinking abilities. The modules cover many topics, including mathematics in daily life, across non-science disciplines, and around everyday environments and experiences. Money management, statistical literacy, and quantitative reasoning were present in every module. Instructors utilized various instructional techniques including cooperative learning, think-pair-share, and read aloud to present the materials to students. S3 Appendix contains a syllabus for the course including all the details about each module delivered.

Note that the pre-test and post-test questions were the same presented in S2 Appendix. Students' scores on the pre-test are unrelated to course activities. Students have not completed any course activity prior to being pre-tested on critical thinking. They were unaware of activities to be completed throughout their course. The pre-test scores were reflections of students' baseline critical thinking abilities.

The mathematics course intervention was designed in light of the empirical support of the Theory of Formal Discipline [33]. This theory predicts that exposure to mathematics enhances students' reasoning, inferential, and logical skills. It is expected that if students complete several cognitively demanding activities requiring higher order reasoning abilities, they will foster better information organizing, explaining, interpreting, argumentation, and communication skills, facets of critical thinking [32].

The course mandated students to prepare independent research using posters. Participants prepared personal budgets using Microsoft Excel. Students completed several game-based learning activities requiring them to identify information, explanations, and arguments. Numerous reflections were inserted helping students identify their assumptions, as well as those of their peers, and critique them. Data collection and analysis projects involving basic statistics were inserted to involve students in doing science activities.

The mathematics course featured several elements and activities that fostered the development of critical thinking skills among participants. Activities like the preparation of posters, data-driven reports, reflections, and personal budgets engaged students actively in analyzing information, organizing them in meaningful ways, and constructing defensible logical arguments.

**Data collection.** Each instructor administered an entrance exam featuring the two questions assessing the critical thinking ability for students at the first meeting of the course. The same test was given at the end of the course to assess the effectiveness of the course on enhancing critical thinking abilities. Each student handed a written in-class paper to their instructors once they completed the assessment. Instructors then graded each question separately using the rubric in S1 Appendix. Each student received five scores per question: explanations of issues, strategy, implementation, conclusion, and the average of the four criteria.

**Ethical considerations.** An Institutional Review Board approval was obtained from Qatar University to conduct this research. Informed consent from each participant was collected in written format. Participants' identifiable information were not collected. Students were not given extra credit nor compensation for their participation. Students retained the choice of not completing the assessment without any repercussions. Instructors' approvals were also obtained via informed consent requiring written signatures. S4 Appendix includes copies of the IRB approval, as well as informed consent forms.

**Data analysis.** Paired-t-tests are used to measure the difference in critical thinking abilities for students before and after completing the mathematics in society course. This hypothesis test is appropriate, given that the two measures are taken for the same variables on the same subjects. It detects whether the means' differences in the five measures of critical thinking were due to the completion of the course or not.

To validate the measurement criteria, several correlation analyses were performed. First, an inter-rater reliability analysis was conducted by correlating the three raters' scores to ensure consistency of grading. Second, to ensure the validity of the critical thinking ability measure, a criterion validity correlation analysis with students' self-reported final grades on Arabic composition, English Composition, and Global Affairs was conducted. Further, the reliability of the two questions was assessed by correlating students' scores across different sections, as well as by internal-consistency measures using Cronbach Alpha and Split-Half reliability metrics.

## The qualitative phase

**Focus groups design.** Focus groups designs are appropriate when researchers desire to collect in-depth information concerning participants' experiences. This research held a focus group with eight students who participated in the experimental study. The focus group featured questions about students' thoughts on the effectiveness of the course in fostering their critical thinking skills. More specifically, students were asked to report their perceptions of the course instructional methods, and their own conceptualizations of their critical thinking progression throughout the course.

**Participants.** Eight students participated in a two-hour long focus group held at Qatar University campus to gather information about their experiences during the course. Four males and four females students participated in the session. Two students were sophomores, three students were juniors, and three students were seniors. Four students were Qatari nationals and four students were non-Qatari citizens. Students' ages ranged between 20 and 23. All participants were speakers of Arabic and reported sufficient fluency levels in English for academic purposes.

**Data collection.** A standard questions protocol was prepared to facilitate the data collection throughout the focus group. The author and two graduate assistants working for the core curriculum unit in the quality assurance department organized the focus group. An invitation to all participating students was sent requesting eight students' participants for the study. Once eight students responded with agreeing to participate in the focus group, the graduate students provided them with details on the location, and timing of the vent. The focus group session took place at a conference room at Qatar University campus after the conclusion of the semester featuring the course in the experimental study. Every participant signed a written informed consent form agreeing to the terms and conditions of the research. Confidentiality was assured, and the IRB committee approved the study. The graduate assistants recorded and transcribed the responses of all eight participants.

Questions on the focus group focused on students' evaluations of their experience. First, students were asked to rate their course using narratives. Second, students were asked about how the course influenced their critical thinking skills. Third, students were asked to cite specific examples and illustrations that helped them achieve better critical thinking abilities. Students were questioned about their opinions of specific instructional methods employed during the course. Finally, students were asked whether they would recommend the course to others, and their reasoning behind their answers.

**Data analysis.** Thematic analysis was used to analyze the responses of the focus group session. Inductive reading of responses informed the construction of themes. Each response was

read on its own and themes were constructed. Then, all responses were read altogether to refine the constructed themes. This step resulted in merging, collapsing, and organizing themes to account for the variation in individual responses. A final set of themes was constructed. The graduate assistants analyzed the information in a similar fashion to ensure the dependability and credibility of results obtained.

## Results

### Instrument validation

Instrument validation is needed to verify if the suggested measures capture the intended meaning. In this case, it is required to confirm if the two questions assessment is sufficient to measure participants' level of critical thinking. Fig 1 demonstrates the validity of the two questions assessment measuring critical thinking in this study. Students completed the short-form of the Glaser-Watson Critical Thinking Appraisal, which significantly correlated with their scores on the assessment administered in this research. Note that students' pre-test scores multiplied by 10 are used to be consistent with the WGCT-SF out of 40. The figure clearly suggests a low critical thinking ability on both tests, and a positive correlation of r = 0.80. Note that Fig 1 displays the correlation between students' scores prior to the completion of the course. A high correlation between two tests is an indication of measuring the same or similar trait. A correlation of 0.80 between the test in S2 Appendix and the WGCT-SF suggests that both tests measure a common construct, and this is critical thinking.

The Watson-Glaser Critical Thinking Appraisal-Short Form has forty possible points that could be converted into percentages or percentiles. The test assesses students abilities on inference, recognition of assumptions, deduction, interpretation, and evaluation of arguments. This set of abilities cover core dimensions of critical thinking including defining problems, identification of important information for finding solutions, recognition of explicit and implicit assumptions, constructing arguments/hypotheses, and drawing logical and empirical conclusions. The test is patented and cannot be reproduced without obtaining a license from the copyrighted institution. In this research, the forty items 1994 English version was translated into Arabic by a panel of higher education faculty at Qatar University. The appraisal is a set of scenarios from common life contexts like schools, workplaces, or organizations. Each scenario is followed by a number of items to be answered by participants. All items have

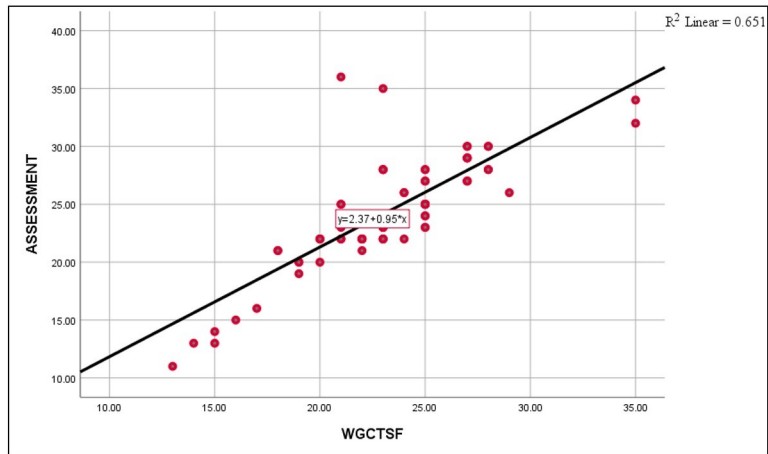

**Fig 1. Watson-Glaser critical thinking short-form and the two questions assessment.**

choices ranging from 2 to 5 for test takers to select from. The total score is the sum of all items on the test. Watson and Glaser calculated the reliability coefficient for this version in 1994 and reported a Cronbach Alpha of 0.81. Note that students only took the WGCT-SF 1994 Arabic translated version prior to completion of the course to estimate the validity of the two questions' instrument in S2 Appendix. Both tests were taken at the same session.

The main proposition of this analysis is that a course in mathematics using instructional strategies fostering critical thinking skills at the college level enhances students' inference, recognition of assumptions, interpretation, strategy formulation, hypotheses construction, and explanation skills. This course is expected to improve students' core critical thinking skills. Therefore, it is hypothesized that exposure to the materials, activities, instructional methods, and assessments in the course lead to gains in critical thinking abilities measured by S2 Appendix questions: explanation, strategy, implementation, and conclusion. Exposure to baseline measurements of critical thinking is thought to have minimal effects on students' critical thinking skills. Students will not be inclined to study the tests they took prior to their course. They will likely complete the expected assignments from them during the to be taken course.

The assessment questions of Qatari students critical thinking skills were validated by a panel of five experts teaching mathematics, English, history, computer science, and biology. All experts possessed ten years of undergraduate teaching experience at various institutions in the Arab World and abroad. The experts rated the questions on their ability to capture four key dimensions of critical thinking: identification of relevant information, synthesizing information in meaningful ways, applying strategic reasoning to address the problems or issues at hand, and surmising informed conclusions based on available evidence. The rating included three categories: 1 = does not measure all four core skills, 2 = measures some of the core skills, and 3 = measures all the skills. All five experts agreed that the questions used to assess critical thinking among Qatari students measured all intended domains of critical thinking giving an average of 3 to both questions.

Since test-retest reliability was not an available option to the researcher given administrative constraints placed by the institution, many measures of reliability have been estimated to ensure the consistency of the instrument. Table 1 displays inter-item correlations between students' average scores on critical thinking in the seven sections that participated in the study. Scores across all sections are strong with correlations exceeding 0.90 pointing to a consistent trend of test reliability. Table 2 displays inter-item correlations for all instructors grading the questions. Associations are strong indicating a consistent grading method adding more reliability to the way raters assessed students. The critical thinking construct internal consistency was assessed by Cronbach Alpha using students' scores on the 4 criteria. Alpha was equal to 0.91 indicating a high level of instrument reliability.

**Table 1. Average score correlations across all sections.**

|  | Section 1 | Section 2 | Section 3 | Section 4 | Section 5 | Section 6 | Section 7 |
|---|---|---|---|---|---|---|---|
| Section 1 | 1.00 |  |  |  |  |  |  |
| Section 2 | 0.92 | 1.00 |  |  |  |  |  |
| Section 3 | 0.93 | 0.94 | 1.00 |  |  |  |  |
| Section 4 | 0.90 | 0.91 | 0.91 | 1.00 |  |  |  |
| Section 5 | 0.94 | 0.92 | 0.93 | 0.91 | 1.00 |  |  |
| Section 6 | 0.93 | 0.94 | 0.92 | 0.93 | 0.93 | 1.00 |  |
| Section 7 | 0.91 | 0.90 | 0.94 | 0.94 | 0.90 | 0.92 | 1.00 |

Cronbach Alpha = 0.91.

**Table 2. Inter-rater reliability.**

|  | Rater 1 | Rater 2 | Rater 3 |
|---|---|---|---|
| Rater 1 | 1.00 |  |  |
| Rater 2 | 0.93 | 1.00 |  |
| Rater 3 | 0.91 | 0.94 | 1.00 |

Table 3 demonstrates high correlations among all measures of critical thinking, as well as the overall summated scale. This indicates a stable structure to the construct, and points to the validity of using the four criteria to assess critical thinking. Table 4 displays correlations between the average critical thinking score, and self-reported students' average score for three courses. Correlations between critical thinking and scores on Arabic, English, and Global Affairs courses are moderate. This confirms the expectation that critical thinking plays an important role in determining students' performance in college establishing the validity of the measure.

Figs 2–5 demonstrate the poor performance of Arab college students prior to the intervention. Regardless of gender, seniority, status, and major. Notice that students scored fair on the four criteria of critical thinking indicative of low critical thinking abilities. Students' scores on explanation, implementation, strategy, and conclusion did not differ much during the pre-measurement phase. The overall average of the four criteria for students was low, fair, prior to the receipt of the critical thinking intervention.

Figs 2–5 displays the improvement of critical thinking abilities among participants after finishing all the activities within the course. Across all groups, students improved significantly raising their critical thinking abilities from fair to good. The gains obtained from the intervention were observed on all criteria, as well as the average score of critical thinking. Students' explanation, implementation, strategy, and conclusion skills rose as a function of their engagement, completion, and effort expensed throughout the course taken.

Demographic variation does not seem to influence critical thinking abilities among Arab students. Gender, seniority, status, and major do not carry significant effects on changes in explanation, implementation, strategy, and conclusion facets of critical thinking. Males, females, expatriate, domestic, STEM, non-STEM, freshman, and seniors all performed similarly during the pre-measurement and post-measurement phases of critical thinking.

Table 5 shows the results of a simple means comparison investigating whether the means on explanation, implementation, strategy, conclusion, and their average differ between the pre-measurement and post-measurement phases interrupted by the critical thinking course intervention. Results indicate that all means are significantly higher during the post-measurement phase compared to the pre-measurement reading of critical thinking abilities. Table 6 presents the pair-t test results formally testing whether the differences observed are statistically significant or not. Results indicate that all differences are statistically significant at the 0.01 level. This indicates that given the data, students scored much higher on critical thinking as a

**Table 3. Internal consistency of criteria for grading.**

|  | Explanation | Strategy | Implementation | Conclusion | Average |
|---|---|---|---|---|---|
| Explanation | 1.00 |  |  |  |  |
| Strategy | 0.93 | 1.00 |  |  |  |
| Implementation | 0.93 | 0.96 | 1.00 |  |  |
| Conclusion | 0.91 | 0.92 | 0.91 | 1.00 |  |
| Average | 0.96 | 0.97 | 0.95 | 0.95 | 1.00 |

**Table 4. Criterion validity.**

|  | Arabic Composition | English Composition | Global Affairs |
|---|---|---|---|
| Explanation | 0.42 | 0.49 | 0.51 |
| Strategy | 0.54 | 0.59 | 0.62 |
| Implementation | 0.44 | 0.48 | 0.54 |
| Conclusion | 0.56 | 0.59 | 0.57 |
| Average | 0.49 | 0.54 | 0.58 |

result of completing a specialized course designed to improve their explanation, implementation, strategy, and conclusion skills.

Table 7 presents the focus groups' thematic analysis results. Notice that four overarching themes define the responses of the eight participants in the focus group. These are: satisfaction, engagement, enjoyment, and relevance. Students highlighted their satisfaction with the course. They indicated that the materials were relevant, engaging, and enjoyable. Students noted the engagement of the instructor, the content, and their peers. Students appreciated the use of real-world applications, autonomous learning, self-directed study, and most important collaborative projects decussating collaborative and deliberative elements.

The thematic analysis from participants' responses on their critical thinking process development confirms earlier theories on critical thinking construction. Students reported low levels of reflection during their first and second weeks in the course. Student two reported "I hardly took anything seriously at the outset of the semester. I did not know how to even read the questions presented'. Once challenge is introduced to students, they begin considering thinking critically. Student six noted "once the instructor started giving us puzzles, mystries, and conundrums, we started coming up with some ideas using the information available to us". Once challenged, students develop basic critical thinking competencies. Student three suggested that "I began making categories of information, classifications of numbers and typologies of items handed or presented to me". Once learners are exposed to efficient ways of critical thinking practice, they become avide users of them. Student eight remarked "when the instructor showed me what a good reesrach question, hypothesis, data, reliability and validity look like in real life, I began applying those ideas in my answers". With the facilitation of the instructor, content, and peers, students become better users of critical thinking. Student five concluded "towards the end of the class, I felt comfortable creating tables, figures, and

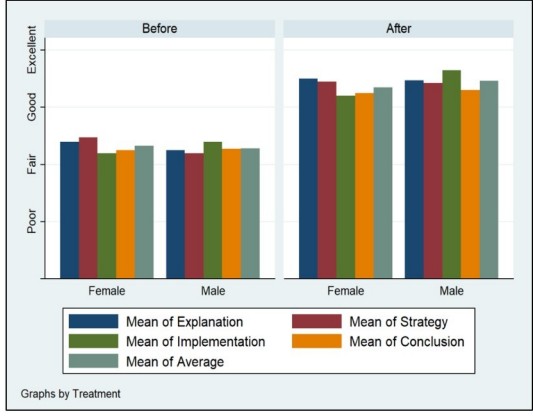

**Fig 2. Gender and critical thinking abilities.**

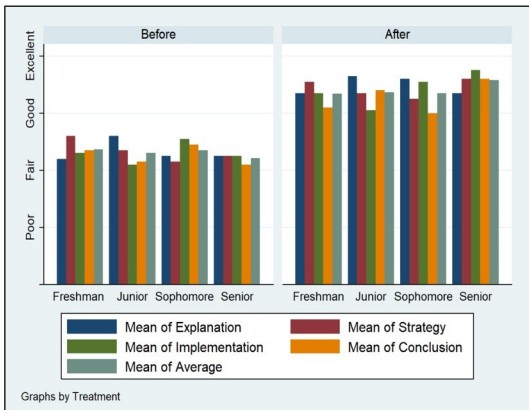

**Fig 3. Seniority and critical thinking abilities.**

illustrations to convey complex information and make simple arguments about them using numbers and pictures". The critical thinking progression process goes through six stages: absent reflection, early challenge, beginning practice, continued and sustained practice, advanced user, and master teacher (thinking development) [28, 29]. Responses of participants provide evidence supporting the proposed stage theory of critical thinking development.

The course has facilitated the development of critical thinking skills and practice among participants in several respects. Students ability to question arguments and talking points has improved over the course of the semester. Participants cited starting the course with minimal questioning practice, and ending the course with high levels of inquisitiveness and curiosity. Participants' problem-solving skills evolved from nascent to satisfactory as a function of completing the course. Many students expressed intimidation, fear, and hesitancy when faced with mathematical or statistical problems.

Focus group responses revealed significant improvements in students' abilities in identifying relevant information, unitizing problems, determining courses of actions, and following set processes to achieve reasonable solutions. Students focused on how the course improved their skills in identifying sound empirical evidence. Participants learned that randomized trails, meta analyses, systematic reviews are superior to cross-sectional and observational studies. They also learned the differences between qualitative and quantitative research, which

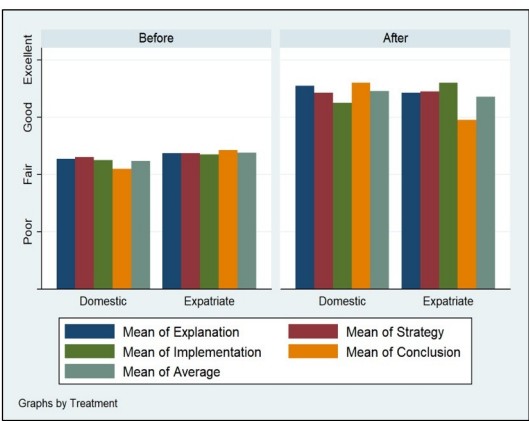

**Fig 4. Status and critical thinking abilities.**

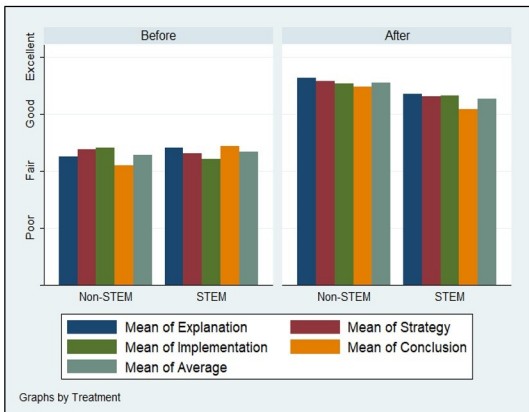

**Fig 5. Major and critical thinking abilities.**

helped them in judging the quality of evidence when they performed independent research throughout the course. Students expressed their self-reflection skills development throughout the course. They began writing their reflections without offering much insight on what they truly believe about anything they write about. By the end of the course, they developed their own voice as a result of learning how to question others' assumptions, as well as ideas. They now have their own active voice when critically appraising others' arguments.

When asked about critical thinking strategies learned throughout the course, participants explicitly referenced information organizing methods, argument construction techniques, and evidence evaluation approaches. Students one, three, and seven all mentioned that classification, categorization, and construction of typologies to organize various information in meaningful ways are new skills they intend to use in other courses. Students two, four, and eight indicated that the use of basic statistics, empirical facts, and numbers in constructing arguments is an essential ability for college success, and one to be practiced by them even after graduation. Students five, six, and eight mentioned their improved abilities to gauge the merit of evidence using quantitative and qualitative research criteria. They emphasized the importance of methodological appraisal prior to taking any finding in a noticeable manner. Students expressed the progression of their skills in thinking critically about every task they face using the above mentioned strategies, as well as others.

**Table 5. Simple means' comparison.**

| | | Mean | N | Std. Deviation | Std. Error Mean |
|---|---|---|---|---|---|
| | | **Paired Samples Statistics** | | | |
| Pair 1 | Pre-Explanation | 2.3250 | 80 | .89690 | .10028 |
| | Post-Explanation | 3.4875 | 80 | .85675 | .09579 |
| Pair 2 | Pre-Strategy | 2.3375 | 80 | .92700 | .10364 |
| | Post-Strategy | 3.4375 | 80 | .89787 | .10038 |
| Pair 3 | Pre-Implementation | 2.3000 | 80 | .90568 | .10126 |
| | Post-Implementation | 3.4250 | 80 | .91090 | .10184 |
| Pair 4 | Pre-Conclusion | 2.2625 | 80 | .95126 | .10635 |
| | Post-Conclusion | 3.2750 | 80 | 1.17973 | .13190 |
| Pair 5 | Pre-Average | 2.3250 | 80 | .88267 | .09869 |
| | Post-Average | 3.4500 | 80 | .89866 | .10047 |

**Table 6. Paired t-test results.**

| | | Paired Differences | | | | | t | df | Sig. (2-tailed) |
|---|---|---|---|---|---|---|---|---|---|
| | | Mean | Std. Deviation | Std. Error Mean | 95% Confidence Interval of the Difference | | | | |
| | | | | | Lower | Upper | | | |
| Pair 1 | Pre-Explanation–Post-Explanation | -1.16250 | .98654 | .11030 | -1.38204 | -.94296 | -10.540 | 79 | .000 |
| Pair 2 | Pre-Strategy–Post-Strategy | -1.10000 | .96259 | .10762 | -1.31421 | -.88579 | -10.221 | 79 | .000 |
| Pair 3 | Pre-Implementation–Post-Implementation | -1.12500 | .98566 | .11020 | -1.34435 | -.90565 | -10.209 | 79 | .000 |
| Pair 4 | Pre-Conclusion1 –Post-Conclusion | -1.01250 | 1.08492 | .12130 | -1.25394 | -.77106 | -8.347 | 79 | .000 |
| Pair 5 | Pre-Average–Post-Average | -1.12500 | .99842 | .11163 | -1.34719 | -.90281 | -10.078 | 79 | .000 |

**Course activities and students' critical thinking skills.** A series of reflection exercises piqued the interest and curiosity of students challenging their non-reflective thinking habits

**Table 7. Students' perceptions of the course.**

| Theme One: Satisfaction | Theme Three: Enjoyment |
|---|---|
| Student Two: "Overall, this course was well organized, and kept me interested. I can say that I would recommend it to my friends because I am satisfied". <br><br> Student Five: "Looking at everything together, this course gave me a nice experience. I was satisfied with the instructor who put me first, and truly listened to what I had to say. The instructor helped me with many activities. Now, I am satisfied with my skills in information identification, organization, and presentation". <br><br> Student Seven: "I am content with the activities we performed throughout the semester. They were challenging. They were satisfying. I had a great deal of conversation with my peers that satisfied my cravings for collaboration and deliberation". | Student Three: "I enjoyed the puzzles. I loved the mysteries. The instructor had an enjoyable style of telling stories. I learned how to make my talk more fun and engaging. This course was not only about math, it was teaching us how to enjoy doing math for life applications". <br><br> Student Six: "The most thing I liked about the course was working with peers. I learned a lot from my classmates. They helped me when I could not understand something. I really enjoyed getting to know people, and I am friends with them now. They help me pick courses and make me a better thinker". <br><br> Student Eight: "Taking all things considered, the class was enjoyable. I liked the mystery games. I needed to find the clues, stick them together, and present a holistic story backed by good argumentation. I want to become a forensic criminal justice professional. This course was about math, but it taught me how to think deeply". |
| **Theme Two: Engagement** | **Theme Four: Relevance** |
| Student One: "There was no single session throughout the semester where I felt inactive. In every class period, the instructor had us work on something. This kept everyone preoccupied and engaged with the materials, activities, and the instructor". <br><br> Student Four: "I was doing a task for most of the time I spent in class. Even at home, I had something to do. The instructor tried his best to keep everyone busy working on something. From my observation, every student around me was engaged either in doing a problem or with a peer to solve a problem". <br><br> Student Eight: "I felt overwhelmed at times during the semester. I was working on a project, puzzle, game, or a mystery. The course kept me working all the time. I worked hard for this course. I truly could say that I was engaged with all my senses". | Student One: "Everything we worked on was from real life. The instructor gave us examples from dealing with vendors to having money deposited in the bank. We tried to interact with real statistical data to formulate hypotheses and questions to be answered. I feel that the instructor made a special effort to make the course materials as real as possible". <br><br> Student Four: "The real-world independent project taught me how to actually examine real-world issues. I practiced how to formulate hypotheses, think of relevant data sources, their reliability, and make arguments based on the". <br><br> Student Six: "I liked the investing examples provided by the instructor. I also enjoyed trying to practice my investment skills using hypothetical scenarios based on real world stories. I wanted to understand how loans work in real-life, and this course offered many puzzles and questions dealing with banking". |

by exercising a modicum of critical reasoning throughout the semester. Every session, students were asked to reflect on the presented topic through small groups' discussions, as well as in writing. For instance, students were asked to provide their opinions as to why mathematics is an important field to their daily lives. Subsequent course sessions asked students to provide real-life examples from their daily interactions about mathematics applications. This series of exercises developed the beginner's critical thinking abilities through challenging preconceptions and considering new ideas about a topic neglected by the thought processes of students.

Students prepared a poster connecting their ideas to convey their view on their selected real-world mathematics application. Throughout this project, students needed to formulate questions, hypotheses, collect information, organize them meaningfully, and present conclusions to classmates. Further, students needed to find reliable sources, as well as statistics to back their arguments. They needed to deliberate with their peers on how to improve their posters, and present the final product to their peers at the conclusion of the term. This project cultivated students' skills in generating relevant inquiries, solve arising problems, build strong independent research skills, and furnish scientific inquiry conceptualization, investigation, and communications abilities.

Students constantly interacted with technology practicing their information gathering, organization, and analysis skills, core domains of critical thinking. Students completed a set of Excel exercises throughout the semester. They were asked to collect data from their classmates, input them into Excel, organize, and analyze them in meaningful ways. Students were asked to share their results with their peers in collaborative settings, and write short reflections on their skills development throughout the process. Students' reflections demonstrated how their skills developed from rudimentary to advanced levels in data collection, entry, screening, and analysis.

Throughout the semester, students were asked to work on real-world mathematics problems, and practice them using mobile games at home. The problems contained a set of information, and inputs where students may locate the relevant material to identify a strategy, implement it, and formulate a conclusion. In each of these exercises, students were asked to identify the set of important information given to find the solution, identify the strategy, explain it in detail, and implement it to formulate a coherent conclusion backed with evidence. In their reflections, students have voiced their readiness to tackle any problem using the identification, strategy formulation, implementation, and conclusion thought process. They exhibited progressive improvements in critical thinking and problem solving core skills in mathematics, as well as other domains throughout the class.

## Discussion

The findings of the current research confirms the well-established conclusion that Arab college students perform poorly on critical thinking assessments. Students received low scores on all four metrics of critical thinking during the pre-measurement phase. This is consistent with the sufficient scoring trend reported elsewhere in the literature [28, 31]. Such low performance is uniform across both genders, majors, and instructors [26, 30, and 32]. By the same token, the findings indicate that a sizable improvement in critical thinking ability is observed once an intervention is administered. Results indicated that students shifted from fair to good once they were exposed to the course that enhanced their critical thinking abilities.

This research does not corroborate the previously held belief that medical, science and technology students possess higher levels of critical thinking skills compared to business, social science, and humanities learners [7, 23, 28, and 30]. The pre-measurement assessment indicated that students performed similarly regardless of their major. Simultaneously, results indicated

that gains in critical thinking are similar across students, and their major did not influence the pattern of enhancements. A potential explanation for this finding is that the sample utilized in the current study is heterogenous, and does not only come from a single group of students like in previous studies. Further, earlier research on medical students included a higher proportion of advanced undergraduates who have completed many courses and are likely to exhibit better critical thinking compared to other groups of students [7, 28].

The present findings corroborate that the combined use of collaborative, game-based, reflective, technology-based, simulation-based learning strategies in mathematics for daily life courses foster universal critical thinking skills. The use of interactive collaborative discussions helps students communicate their arguments in a better fashion while appraising their peers' arguments more critically [26, 30, and 31]. Game-based and simulation-technology mediated learning help students become better at identifying useful information, organizing data, and better analysts of information. They also allow students to construct their own readings and understandings of the information generating creative tables, graphs, conceptual maps, and new ideas. Reflections permit students to question their own assumptions, as well as those of others. They enable students to present their ideas backed with their attempts of making cogent arguments. Amalgamating all such teaching methods while teaching mathematical and statistical literacy facilitate the development of core dimensions of critical thinking skills [7, 33].

This research does not find support to the hypothesis that expatriate students perform better than their domestic peers from the Arabian Gulf region. The pre-measurement phase demonstrated low critical thinking abilities for both expatriate and Qatari students. Further, the improvements on critical thinking are similar across both groups. One possible reason explaining this observation is the low sampling heterogeneity in previous studies that featured small samples of expatriate students. Further, this study featured a large sample of both domestic, as well as expatriate students that vary across many conditions like major, gender, and class standing.

The findings of this study lent support to the no difference hypothesis between the two genders and critical thinking [11, 13, and 29]. Across all five measures of critical thinking during both the pre-measurement, as well as the post-measurement phases, males and females performed similarly with no practical difference. While this conclusion contradicts the more confirmed inference that females outperform males in college, it is explained by the similar exposure of both genders to the same type of education, and college experience. Qatar University admits a competitive pool of students, and this could make up for the males' performance gap observed when college achievement is investigated.

The study found no support for the seniority hypothesis and critical thinking. Across all five measures, seniors performed similarly to other students' groups on the pre-measurement, and post-measurement phases. This findings could be the result of the highly competitive pool of students at Qatar University. Another plausible reason is that previous studies did not feature large numbers of students across different groups. They favored a higher proportion for seniors compared to freshman, sophomore, or junior.

Mixed-methods research design provided an ideal tool to pursue this research for three particular reasons. First, it allowed for a comparison of quantitative results and quantitative findings. Second, the qualitative component, in particular, enabled the researcher to reflect participants' point of view. Last but not the least, it encourages more scholarly engagement on the topic by inspiring methodological flexibility.

## Recommendations

Arab colleges and universities are urged to implement critical thinking interventions. All students are expected to master critical thinking by the time of their graduation; however, this is

not achieved. One of the solutions is to expose students to at least one specialized course or more that target the core skills of critical thinking. This research has found support to the proposed intervention implemented at Qatar University. Such an intervention could be implemented in other Arab higher education institutions.

Arab colleges and universities need to prioritize critical thinking assessment. Few institutions in the region irregularly evaluate critical thinking, and the vast majority do not. This does not help institutions in achieving their highest potential. Instructors, department chairs, college deans, and university administrators need to aggressively require students to complete critical thinking assessments. Baseline evaluations of critical thinking help educators plan better for the improvement of critical thinking. Students could complete an overall assessment at the beginning of their college experience, as well as at the end of it.

The improvement in critical thinking at the student requires a change of instructional approaches. Instructors need to incorporate more conducive practices that foster the learning and mastery of critical thinking. Those include collaborative, cooperative, active, independent, and problem-inquiry learning approaches. The incorporation of technology like simulations, animations, and gaming is also recommended for students to develop better critical thinking skills. Besides, game theoretical methods can be used together with the statistical methods to address the effectiveness and the outcomes of the strategies [23, 24]. Furthermore, increasing instructor-student interaction also increases the likelihood of students' investment in learning, which improves critical thinking mastery.

## Limitations

The failure to use a classical experimental design with a randomized control approach limits the quality of the research. While the current design controlled for many confounders like gender, seniority, and major, it did not control for everything that could potentially explain critical thinking skills changes among students. Further, the lack of a control group limits the researchers' ability to generalize the findings observed by the quasi-experimental study performed.

The difficulty in recruiting a large representative random sample limits the study's external validity. While the sample included in the study is heterogenous representing many different segments of the population, its selection was not random nor large. The use of 80 students in a quasi-experimental design appears sufficient, however, the great variety of students in the Arab World requires a larger sample from each group in the population. In addition, Qatar University students come from highly qualified pool of college students in the Arab World.

## Future research

An increased investment in the development of Arab critical thinking assessments is warranted. Most researchers and institutions rely on Western tools that are replete with references foreign to Arab college students. Universities, and research centers along with future researchers are expected to design, validate, and implement original evaluations in their campuses. This yields a more authentic assessment of critical thinking without the reliance on tools developed for foreign populations.

Large-scale, as well as longitudinal studies should be carried to investigate the differences in critical thinking skills among Arab college students. Large representative samples that are randomly designed should be the focus of future research. Similarly, tracking the changes within critical thinking overtime and discovering the factors behind such changes should be another active area of research.

## Conclusions

Critical thinking is an indispensable skill for college success. It provides learners with a keen ability to reach logical inferences based on an identified strategy implemented through appropriate understanding of information. Such a versatile ability makes individuals more likely achieve better career prospects, business aspirations, and creative works. Critical thinking like other skills could be acquired through specialized learning. This investigation lends support to the Theory of Formal Discipline arguing that mathematics for daily life instruction improves students' universal critical thinking skills.

The analysis presented a quasi-experimental investigation evaluating the effectiveness of a quantitative reasoning course on Arab students' critical thinking abilities at Qatar University. After completing the course, students' average critical thinking scores improved from fair to good. The gains in critical thinking skills were uniform across its four dimensions: explanation of information, strategy identification, implementation of solutions, and formulating logical conclusions. Demographic and class characteristics did not impact the positive significant impact of the quantitative reasoning course on students' abilities.

The findings lend support to the use of active learning, collaborative, and innovative technology in teaching at the college level. Students demonstrated more interest and exhibited more effort in learning once think-pair-share, small group activities, and conceptual mapping were introduced. Further, cooperative, and collaborative learning increased peer-to-peer interactivity, which motivated stunted learners to excel like their groups' leaders. The use of simulations, games, animations, and real-world projects requiring data collection, as well as analysis motivated students to practice learning on their own generating better environments for critical thinking cultivation. The experiment described in the present manuscript is flexible, scalable, and easily replicated in similar contexts in the Middle East.

## Supporting information

**S1 Appendix. CCP-SLO (5): Think critically and creatively in a variety of methods in order to make decisions and solve problems.**
(DOCX)

**S2 Appendix. Translation of the two questions instrument.**
(DOCX)

**S3 Appendix. The special course syllabus.**
(DOCX)

**S4 Appendix. Informed consent.**
(DOCX)

**S1 Data.**
(XLSM)

## Author Contributions

**Conceptualization:** Wael Yousef.

**Data curation:** Wael Yousef.

**Formal analysis:** Wael Yousef.

**Investigation:** Wael Yousef.

**Methodology:** Wael Yousef.

**Project administration:** Wael Yousef.

**Supervision:** Wael Yousef.

**Validation:** Wael Yousef.

**Visualization:** Wael Yousef.

**Writing – original draft:** Wael Yousef.

**Writing – review & editing:** Wael Yousef.

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
