## [Editor Report · Decision Letter 0]

8 Apr 2021

PONE-D-21-09116

An Assessment of Critical Thinking in the Middle East: Evaluating the Effectiveness of Special Courses Interventions

PLOS ONE

Dear Dr. Wael Yousel,

Thank you for submitting your manuscript to PLOS ONE. After careful consideration, we feel that it has merit but does not fully meet PLOS ONE’s publication criteria as it currently stands. Therefore, we invite you to submit a revised version of the manuscript that addresses the points raised during the review process.

1. The topic addressed in this manuscript is relevant and permeates current discussion on ways to prepare students to face labor, economic, and social changes. Although the manuscript’s focus is on ways to assess critical thinking, both the introduction and literature review sections include statements that refer to different ways to conceptualize the same theme (critical thinking) and to better situate the study it seems important to include a clear characterization of what conceptualization is endorsed in this research. In particular, the discussion of students’ development of critical thinking across different domains and its relation to the study of specific domain such as mathematics.<o:p></o:p>

2. The section “assessment of critical thinking skills among Arab College students” is tedious and offers little information on what those assessments really mean in terms of students’ critical thinking behaviors. This section could be reduced and restructured to address and discuss what those students’ results from different tests (and versions) contributed to actually frame the study. In particular, the extent to which these test results helped to structure and implement the course.<o:p></o:p>

3. It the section “instructional methods fostering critical thinking”, it is not clear how the review of this section really contributed to frame the course that was used to assess the students’ critical thinking. Indeed, it seems important to document the extent to which issues such as non-traditional instructional methods, game-based learning, independent and self-regulated learning, etc. were considered during the development of the course sessions.<o:p></o:p>

4. The research design is weak, and it might be improved if a qualitative approach is added or considered to analyze the instructional approach that supported the course activities and the participants’ thought process. The questions to assess students’ critical thinking are not robust enough to actually capture what a critical thinking involves. Showing the corresponding “instrument validation” through the positive correlation of test hides the complexity involved in capture what is essential in students’ critical thinking. In addition, since the sample of the participants was not random, the statistic treatment is questionable. Here, specific examples of how the course activities were implemented and what problems the students worked during the sessions could provide more solid information regarding their problem-solving behaviors including critical thinking. Indeed, what the author shows in the Results section needs to be contextualized in terms of connecting what happened during the course sessions and ways they approaches the two questions and more problems. 

We look forward to receiving your revised manuscript.

Kind regards,

Manuel Santos-Trigo, PhD

Academic Editor

PLOS ONE

Journal Requirements:

Additional Editor Comments (if provided):

1. The topic addressed in this manuscript is relevant and permeates current discussion on ways to prepare students to face labor, economic, and social changes. Although the manuscript’s focus is on ways to assess critical thinking, both the introduction and literature review sections include statements that refer to different ways to conceptualize the same theme (critical thinking) and to better situate the study it seems important to include a clear characterization of what conceptualization is endorsed in this research. In particular, the discussion of students’ development of critical thinking across different domains and its relation to the study of specific domain such as mathematics. Mathematics is the subject used to design and implement a course for students to develop critical thinking.

2. The section “assessment of critical thinking skills among Arab College students” is tedious and offers little information on what those assessments really mean in terms of students’ critical thinking behaviors. This section could be reduced and restructured to address and discuss what those students’ results from different tests (and versions) contributed to actually frame the study. In particular, the extent to which these test results helped to structure and implement the course.

3. It the section “instructional methods fostering critical thinking”, it is not clear how the review of this section really contributed to frame the course that was used to assess the students’ critical thinking. Indeed, it seems important to document the extent to which issues such as non-traditional instructional methods, game-based learning, independent and self-regulated learning, etc. were considered during the development of the course sessions.

4. The research design is weak, and it might be improved if a qualitative approach is added or considered to analyze the instructional approach that supported the course activities and the participants’ thought process. The questions (proposed in the questionnaire) to assess students’ critical thinking are not robust enough to actually capture what a critical thinking involves. Showing the corresponding “instrument validation” through the positive correlation of test hides the complexity involved in capture what is essential in students’ critical thinking. In addition, since the sample of the participants was not random, the statistic treatment is questionable. Here, specific examples of how the course activities were implemented and what problems the students worked during the sessions could provide more solid information regarding their problem-solving behaviors including critical thinking. Indeed, what the author shows in the Results section needs to be contextualized in terms of connecting what happened during the course sessions and ways the participants approached the two questions and more problems. It is not clear what the others three questions were included in the questionnaire since there is little discussion of the students' corresponding answers. Similarly, it is not clear what the participants' interviews contributed to the data analysis and results.

5. In summary, a more consistent and robust version of the manuscript is needed in order to send it to reviewers.
---

## [Author Response · Author response to Decision Letter 0]

30 Apr 2021

Dear Editor, I have revised the paper based on the reviewers` comments.

---

## [Decision Letter · Decision Letter 1]

7 Jul 2021

PONE-D-21-09116R1

An Assessment of Critical Thinking in the Middle East: Evaluating the Effectiveness of Special Courses Interventions

PLOS ONE

Dear Dr. Yousef,

Thank you for submitting your manuscript to PLOS ONE. After careful consideration, we feel that it has merit but does not fully meet PLOS ONE’s publication criteria as it currently stands. Therefore, we invite you to submit a revised version of the manuscript that addresses the points raised during the review process.

We look forward to receiving your revised manuscript.

Kind regards,

Manuel Santos-Trigo, PhD

Academic Editor

PLOS ONE

Additional Editor Comments (if provided):

Although the author sent a revised version of the initial submission, there are still several issues that reviewers pointed out during the reading of the manuscript that need to be revised and supported to get a coherent manuscript. First, the introduction and literature review parts includes several statements linked to equal number of references; but it is not clear what critical thinking position the author actually endorses and rely on framing the study. In particular, it lacks information regarding how the list of elements that supposedly were part of the course was actually incorporated in the course design including the type of activities that were considered to foster the participants development of critical thinking behaviors. Secondly, there is no information regarding the pre-test questions and its possible connection with the activities that were implemented during the course. Indeed, the two questions seem limited to actually assess the participants' critical thinking competencies. The qualitative part results offer little information about the participants' actual answers and their relation to their critical approach to deal with them. The qualitative part focused mainly on the participants perceptions about the course without exploring their ways of thinking to deal with those questions.

Reviewers' comments:

Reviewer's Responses to Questions

**Comments to the Author**

1. If the authors have adequately addressed your comments raised in a previous round of review and you feel that this manuscript is now acceptable for publication, you may indicate that here to bypass the “Comments to the Author” section, enter your conflict of interest statement in the “Confidential to Editor” section, and submit your "Accept" recommendation.

Reviewer #1: All comments have been addressed

Reviewer #2: (No Response)

2. Is the manuscript technically sound, and do the data support the conclusions?

Reviewer #1: Partly

Reviewer #2: Yes

3. Has the statistical analysis been performed appropriately and rigorously? 

Reviewer #1: Yes

Reviewer #2: I Don't Know

4. Have the authors made all data underlying the findings in their manuscript fully available?

Reviewer #1: Yes

Reviewer #2: Yes

5. Is the manuscript presented in an intelligible fashion and written in standard English?

Reviewer #1: Yes

Reviewer #2: Yes

6. Review Comments to the Author

Reviewer #1: I found the addition of Qualitative section does not necessarily strengthen the paper, while it may be good to include them as supporting information.

Design weakness cannot be handled with re-writing.

Reviewer #2: The study investigates the impact of a novel course structure on critical thinking in a mathematics class at Qatar University. The novel course structure aims to target critical thinking skills in particular. The participants initially have a low baseline of mathematical and critical thinking achievement. After a good Introduction and Literature Review, I was left wanting to understand the study thoroughly.

Comment #1:

A significant number of issues have arisen with regards to the assumption that mathematics and Critical Thinking (CT) are aligned, and this discussion is missing in light of how a mathematics course has been targeted specifically for this investigation. [C.f. Inglis M, Attridge N. Does mathematical study develop logical thinking? Testing the theory of formal discipline. London: World Scientific Publishing Europe Ltd; 2016.] There needs to be a true clarity between the mathematical course content and assessment, and then the teaching style and focus on the emerging CT skills. Any link between these two themes needs to be carefully managed. The last sentence of the first paragraph of the conclusion is particularly confusing. Is the suggestion that it was the quantitative reasoning-based nature of the course that contributed to the fostering of the CT skills or the method of instruction that included game-based learning, collaborative and independent learning, real-world projects, etc.?

Comment #2:

Comment #1 feeds into some aspects of the research that would be valuable, if not necessary, to unpack. The Two Questions Instrument in Appendix B are financial mathematics questions. How were these unpacked as CT questions? Having some examples would be worthwhile here. Table 7 provides subjective student perceptions. A Table considering mathematical answers that present more than mathematics but the CT skills therein would be equally appropriate. How were the answers explicitly analysed vis-a-vis CT? How did the five measures get mapped out explicitly in these financial mathematics questions? This would be highly valuable in the context of the murky understanding of CT definitions, how and whether it can be taught or fostered and the issues that have presented themselves in understanding these with comparison to mathematical thinking.

Comment #3:

Appendix A is important; it is also linked to Comment #2. What is “the most efficient solution strategy”? One that is quick or one that reveals an underlying framework that may facilitate further solutions in other domains? There is an underlying presumption of qualities in Appendix A that seems aligned precisely with the difficulties underlying our current understanding of CT. Showing a solution considered most efficient, versus non-efficient, versus acceptable, versus non connected, as the table partitions, would be valuable to see.

Comment #4:

What was the short-form Glaser-Watson test used? Can this be explicitly given and its choice corroborated? Was it the exact same questions (both Glaser-Watson and the 2 financial mathematics) used at the beginning and end of the course? If so, how can we explain an improvement as an improvement in CT and not simply learning the course content or even re-exposure? Could this be unpacked.

Does Figure 1 include data gathered at the beginning and end of course? How does it demonstrate the validity of CT assessment and not simply an education in financial mathematics? Further details could be explored here.

Comment #5:

How did the present authors converge on their particular method to foster CT skills? Here lies the strength of the work. It is not clear at all how the study mentioned in [41] recommends puzzles, mysteries, and conundrums; and study [40] pertains to nursing students. Maybe this section could be shortened? Maybe the authors could celebrate their creativity here more? Giving details to frame the mathematics course as to how it was to build CT skills is of interest (while bearing in mind that how to build CT skills is still an open question).

Comment #6:

Less seriously, the manuscript places a lot of emphasis on the Arabic world, without then truly unpacking why the Arabic world is so different to other communities. The study in question certainly does not present a global analysis. Weak mathematical students are prevalent worldwide, presenting similar issues. Are CT behaviours celebrated anywhere? The opening sentence [line 117] is a strong one. If there is a lack, this suggests a comparison. And what is “sufficient”? Line 131: What is an “unacceptable skill”?

Comment #7:

Errors pertaining to typography/grammar/repetition.

Line 103/104: How is the third area of disagreement different from the first?

Line 155: The wording “proposed” course is confusing because it is no longer proposed but has occurred. This was confusing throughout the document.

A number of typographical errors still appear across the manuscript. It should be carefully proof edited. Here are some: Line 138 “fails”? Line 157 “features”? Line 147: extra space. Line 171 “course”? Line 178 space before reference. Line 211: It’s the CT abilities of students at a university not the CT skills of the university? Line 213: thoughts processes progression? Line 255: “around s’ everyday”? Line 310 “vent”? Line 519: “finding”? Line 568: carried out? Line 571: “over time”? Line 579: “to achieve”? Line 708: bold 37. Appendix A Implementation-Poor: “Demonstrates”?

7. PLOS authors have the option to publish the peer review history of their article (what does this mean?). If published, this will include your full peer review and any attached files.

Reviewer #1: No

Reviewer #2: No

---

## [Author Response · Author response to Decision Letter 1]

28 Jul 2021

Thank you for the opportunity to revise my paper. The reviewers’ comments were both helpful and insightful. I have made a serious effort to address all of the issues raised by them and think that the paper is stronger as a result. In my letter "Response to reviewers", I explained in detail detail what their issues were, and how I attempted to address them

---

## [Decision Letter · Decision Letter 2]

25 Aug 2021

PONE-D-21-09116R2

An Assessment of Critical Thinking in the Middle East: Evaluating the Effectiveness of Special Courses Interventions

PLOS ONE

Dear Dr. Yousef,

Thank you for submitting your manuscript to PLOS ONE. After careful consideration, we feel that it has merit but does not fully meet PLOS ONE’s publication criteria as it currently stands. Therefore, we invite you to submit a revised version of the manuscript that addresses the points raised during the review process.

We look forward to receiving your revised manuscript.

Kind regards,

Manuel Santos-Trigo, PhD

Academic Editor

PLOS ONE

Journal Requirements:

Additional Editor Comments (if provided):

Dear author, your revised version of your manuscript has been reviewed and reviewers agree that you have improved the content of the paper and you have answered some concerns and observations that were made during previous reviews; however, they also have identified some shortcomings that you need to address in the next version of the manuscript:

1. Please present a clean version without highlighting the changes and reduce the length of the paper to 60 000 characters including space.

2. In the introduction and literature review parts, include only relevant papers that were important to support and contributed to frame the study. Avoid presenting a lists of several paper to support a single statement.

3. Review and clarify how Arab College students get assessed differently in comparison with international students, one of the reviewers asked for this clarification

4. Instead of adding texts or contents to previous sections of the old version, please review the content of sections that were asked to clarify and integrate the ideas to provide a coherent argument.

5. Conceptualize and support the choice of only two questions to assess the participants' critical thinking and related the students' work to their course behavior while working on instructional tasks.

6. Avoid long tables, select relevant information and results that were used to answer research questions

7. Focus on discussing the meaning associated with the process of instrument validation, rather than presenting results from pretest, etc.

8. Discuss the extent to which both qualitative and quantitative analysis really contributed to answer research questions and the type of limitations that were faced during the development of the study.

Reviewers' comments:

Reviewer's Responses to Questions

**Comments to the Author**

1. If the authors have adequately addressed your comments raised in a previous round of review and you feel that this manuscript is now acceptable for publication, you may indicate that here to bypass the “Comments to the Author” section, enter your conflict of interest statement in the “Confidential to Editor” section, and submit your "Accept" recommendation.

Reviewer #1: All comments have been addressed

2. Is the manuscript technically sound, and do the data support the conclusions?

Reviewer #1: Yes

3. Has the statistical analysis been performed appropriately and rigorously? 

Reviewer #1: Yes

4. Have the authors made all data underlying the findings in their manuscript fully available?

Reviewer #1: Yes

5. Is the manuscript presented in an intelligible fashion and written in standard English?

Reviewer #1: Yes

6. Review Comments to the Author

Reviewer #1: (No Response)

7. PLOS authors have the option to publish the peer review history of their article (what does this mean?). If published, this will include your full peer review and any attached files.

Reviewer #1: No

---

## [Author Response · Author response to Decision Letter 2]

27 Sep 2021

Response to Reviewers

Thank you for the opportunity to revise my paper entitled “An Assessment of Critical Thinking in the Middle East: Evaluating the Effectiveness of Special Courses Interventions” The reviewers’ comments were both helpful and insightful. I have made a serious effort to address all of the issues raised by them and think that the paper is stronger as a result. I are going to detail here what their issues were, and how I attempted to address them. 

1. Please present a clean version without highlighting the changes and reduce the length of the paper to 60 000 characters including space.

I have reduced the manuscript from 100,000+ characters to less than 60000 characters (without appendices).

2. In the introduction and literature review parts, include only relevant papers that were important to support and contributed to frame the study. Avoid presenting a lists of several paper to support a single statement.

Unnecessary references were removed from the text. 

3. Review and clarify how Arab College students get assessed differently in comparison with international students, one of the reviewers asked for this clarification

This was discussed on page 4-5, by adding the following text: 

“There are several systematic differences between the assessment of Arab students on critical thinking compared to many students’ populations like those of North America, Europe, and East Asia. First, Arab students are exposed to less frequent, as well as systematic critical thinking appraisals. Second, Arab colleges and universities seldom evaluate their students’ abilities in critical thinking. Third, courses at both the undergraduate and graduate levels in Arab colleges and universities do not feature high levels of cognitively demanding tasks like intensive writing, experiential projects, in-class simulations, or independent research. Fourth, Arab college students have performed systematically lower on critical thinking assessments compared to their counterparts around the world. Fifth, Arab students are assessed using translated non-authentically developed instruments within their own environments leading to biased understandings of several items on critical thinking appraisals.” 

4. Instead of adding texts or contents to previous sections of the old version, please review the content of sections that were asked to clarify and integrate the ideas to provide a coherent argument.

The added material in the last review were incorporated to existing text, and unnecessary text was removed. 

5. Conceptualize and support the choice of only two questions to assess the participants' critical thinking and related the students' work to their course behavior while working on instructional tasks.

This was discussed on page 9, by adding the following text:

“The two questions’ instrument intended to measure four fundamental dimensions of critical thinking ability: identification of relevant information, the formulation of a strategy, the implementation of the chosen strategy, and inferring an interpretable statement based on the evidence available. Each of the question requires learners to engage in all four conceptual cornerstones of critical thinking. The use of two questions allows the assessment of critical thinking in a rapid manner without overburdening students with taking hours-long assessments. Correlations between students’ scores on the two questions and a popular appraisal yielded high associations signaling the validity of the two questions’ instrument.”

6. Avoid long tables, select relevant information and results that were used to answer research questions

Suggestions of summarizing the long tables in few sentences were incorporated. The long tables were removed from the manuscript and the appendix to ease the readability of the manuscript on interested readers

7. Focus on discussing the meaning associated with the process of instrument validation, rather than presenting results from pretest, etc.

This was addressed on page 14, by adding the following text:

“Instrument validation is needed to verify if the suggested measures capture the intended meaning. In this case, it is required to confirm if the two questions assessment is sufficient to measure participants’ level of critical thinking.”

8. Discuss the extent to which both qualitative and quantitative analysis really contributed to answer research questions and the type of limitations that were faced during the development of the study

This was addressed on pages 29 and 31, by adding the following texts:

“Mixed-methods research design provided an ideal tool to pursue this research for three particular reasons. First, it allowed for a comparison of quantitative results and quantitative findings. Second, the qualitative component, in particular, enabled the researcher to reflect participants’ point of view. Last but not the least, it encourages more scholarly engagement on the topic by inspiring methodological flexibility.” On page 29

“The difficulty in recruiting a large representative random sample limits the study’s external validity. While the sample included in the study is heterogenous representing many different segments of the population, its selection was not random nor large. The use of 80 students in a quasi-experimental design appears sufficient, however, the great variety of students in the Arab World requires a larger sample from each group in the population. In addition, Qatar University students come from highly qualified pool of college students in the Arab World.” On page 31

---

## [Editor Report · Decision Letter 3]

12 Oct 2021

PONE-D-21-09116R3An Assessment of Critical Thinking in the Middle East: Evaluating the Effectiveness of Special Courses InterventionsPLOS ONE

Dear Dr. Yousef,

Thank you for submitting your manuscript to PLOS ONE. After careful consideration, we feel that it has merit but does not fully meet PLOS ONE’s publication criteria as it currently stands. Therefore, we invite you to submit a revised version of the manuscript that addresses the points raised during the review process.

We look forward to receiving your revised manuscript.

Kind regards,

Manuel Santos-Trigo, PhD

Academic Editor

PLOS ONE

Journal Requirements:

Additional Editor Comments (if provided):

The author has answered comments and suggestions provided by reviewers. In a throughout reading of this version, the author made four main claims that not necessarily are well supported. First, there is no clear evidence that the study provides a systematics assessment of Qatari students in critical thinking abilities. Here, the main assessment used in the study involves two questions that limit to actually explore the participants' thinking behaviors and using the same questions in both pre and post tests assessments reduces the generalization of the results. Second, the author claims that the study provides an easy replication of the study to similar contexts. Indeed, this part still needs to be explained in terms of what mathematics tasks and instructional activities were actually implemented during the intervention part. What is reported in this section only includes general course description without discussing in detail what the content was actually involved in the course sessions and its implementation. Third, a major weakness of the study is the sample involved in the quantitative part, since it did not include a random choice of the participants. Thus, tables 1-6 offers confusing data regarding the instrument validity. Clearly, focusing on two tasks to assess the complexity involved in students' development of critical thinking offers serious methodological limitations to document the participants' critical thinking behaviors. Here, author should make explicit what trustworthiness criteria was used to support the qualitative analysis. And fourth, the results does not address explicitly the way the participants collaborated during the development of the study during the problem-solving sessions. Thus, there is a need to support this fourth claim.

A recommendation here is that the author needs to moderate the language to make those claims and take into account only what the study actually reports including its methodological and results limitations.
---

## [Author Response · Author response to Decision Letter 3]

26 Nov 2021

Response to Editor

Thank you for taking time and paying attention to the manuscript submitted for publication to your journal. I have moderated the language reflecting the limitations of: 

1. Assessment Instrument 

2. Sampling Strategy

3. Difficulty in Replication 

4. The extent of participants' collaboration. 

Please note that this manuscript has gone through three full reviews, and many tables inserted within the manuscript were demanded by reviewers. For instance, the Qualitative section tables were requested by a reviewer in the first round. 

I am sincerely thankful for your time and attention given to the manuscript. Your comments have made the manuscript shorter, concise, and organized. All these qualities are great for readers! 

Thank you

---

## [Editor Report · Decision Letter 4]

19 Dec 2021

An Assessment of Critical Thinking in the Middle East: Evaluating the Effectiveness of Special Courses Interventions

PONE-D-21-09116R4

Dear Dr. Yousef,

We’re pleased to inform you that your manuscript has been judged scientifically suitable for publication and will be formally accepted for publication once it meets all outstanding technical requirements.

Kind regards,

Manuel Santos-Trigo, PhD

Academic Editor

PLOS ONE

Additional Editor Comments (optional):

The author has answered comments, suggestions, and observations in this versiono of the manuscript. Thus, the manuscript is ready to go to the next phase for its publication.
---

## [Editor Report · Acceptance letter]

22 Dec 2021

PONE-D-21-09116R4 

An Assessment of Critical Thinking in the Middle East: Evaluating the Effectiveness of Special Courses Interventions 

Dear Dr. Yousef:

I'm pleased to inform you that your manuscript has been deemed suitable for publication in PLOS ONE. Congratulations! Your manuscript is now with our production department. 

Kind regards, 

on behalf of

Dr. Manuel Santos-Trigo 

Academic Editor

PLOS ONE